# The inside scoop: Comparative genomics of two intranuclear bacteria, "*Candidatus* Berkiella cookevillensis" and "*Candidatus* Berkiella aquae"

Destaalem T. Kidane[1,2], Yohannes T. Mehari[3], Forest C. Rice[2], Brock A. Arivett[4], John H. Gunderson[5], Anthony L. Farone[1,2], Mary B. Farone[1,2]*

1 Molecular Biosciences Program, Middle Tennessee State University, Murfreesboro, TN, United States of America, 2 Department of Biology, Middle Tennessee State University, Murfreesboro, TN, United States of America, 3 Department of Biological Sciences, Auburn University, Auburn, AL, United States of America, 4 Division of Infectious Disease, Department of Medicine, University of Alabama at Birmingham, Birmingham, AL, United States of America, 5 Department of Biology, Tennessee Technological University, Cookeville, TN, United States of America

* mary.farone@mtsu.edu

**Data Availability Statement:** Genome sequence files of "Ca. B. cookevillensis" and "Ca. B. aquae" have been deposited at GenBank database under

## Abstract

"*Candidatus* Berkiella cookevillensis" (strain CC99) and "*Candidatus* Berkiella aquae" (strain HT99), belonging to the Coxiellaceae family, are gram-negative bacteria isolated from amoebae in biofilms present in human-constructed water systems. Both bacteria are obligately intracellular, requiring host cells for growth and replication. The intracellular bacteria-containing vacuoles of both bacteria closely associate with or enter the nuclei of their host cells. In this study, we analyzed the genome sequences of CC99 and HT99 to better understand their biology and intracellular lifestyles. The CC99 genome has a size of 2.9Mb (37.9% GC) and contains 2,651 protein-encoding genes (PEGs) while the HT99 genome has a size of 3.6Mb (39.4% GC) and contains 3,238 PEGs. Both bacteria encode high proportions of hypothetical proteins (CC99: 46.5%; HT99: 51.3%). The central metabolic pathways of both bacteria appear largely intact. Genes for enzymes involved in the glycolytic pathway, the non-oxidative branch of the phosphate pathway, the tricarboxylic acid pathway, and the respiratory chain were present. Both bacteria, however, are missing genes for the synthesis of several amino acids, suggesting reliance on their host for amino acids and intermediates. Genes for type I and type IV (*dot/icm*) secretion systems as well as type IV pili were identified in both bacteria. Moreover, both bacteria contain genes encoding large numbers of putative effector proteins, including several with eukaryotic-like domains such as, ankyrin repeats, tetratricopeptide repeats, and leucine-rich repeats, characteristic of other intracellular bacteria.

## Introduction

Free-living amoebae, found in natural and human-made aquatic environments, are predators of many bacteria, and thus play an important role in controlling microbial populations in the

accession numbers LKHV00000000 and LKAJ00000000, respectively. This project has been deposited at NCBI under the BioProject accession number PRJNA289553.

**Funding:** This work was supported by the National Institute of General Medical Sciences at the National Institutes of Health (https://nigms.nih.gov/) grant R15GM131357-01 to MBF. The funders had no role in study design, data collection and analysis, decision to publish, or preparation of the manuscript.

**Competing interests:** The authors have declared that no competing interests exist.

environment [1]. Some bacteria, however, have evolved to become resistant to predation by these protozoa. These amoeba resistant bacteria (ARB) have mechanisms that not only allow them to survive internalization and digestion by amoebae, but also allow them to replicate within the intra-amoebal environment [2, 3]. Both facultative and obligate intracellular ARB belonging to several evolutionary lineages, including alphaproteobacteria [4–6], betaproteobacteria [7], gammaproteobacteria [8, 9], Bacteroidetes [10] and Chlamydiae [11, 12] have been recovered from free-living amoebae. Among those identified, several ARB such as *Legionella pneumophila* and *Mycobacterium avium* are established human pathogens [13, 14] while others, including *Parachlamydia acanthamoebae* and the Legionella-like amoebal pathogens (LLAPs), have been designated as potential emerging human pathogens [15, 16].

"*Candidatus* Berkiella cookevillensis" (type strain CC99) and "*Candidatus* Berkiella aquae" (type strain HT99) are two ARB that were each isolated from an amoeba present in biofilm recovered from a hospital cooling tower and an outdoor hot tub spa, respectively [17]. Both bacteria are non-spore-forming, motile, obligate intracellular gram-negative bacteria. CC99 bacteria are coccoid shaped with diameters ranging from 0.30 to 0.60 μm whereas HT99 are coccobacilli with a width ranging from 0.30–0.55μm and length ranging from 0.45–0.65μm (Fig 1) [17]. Based on 16S rRNA gene phylogenetic analyses and cellular fatty acid composition

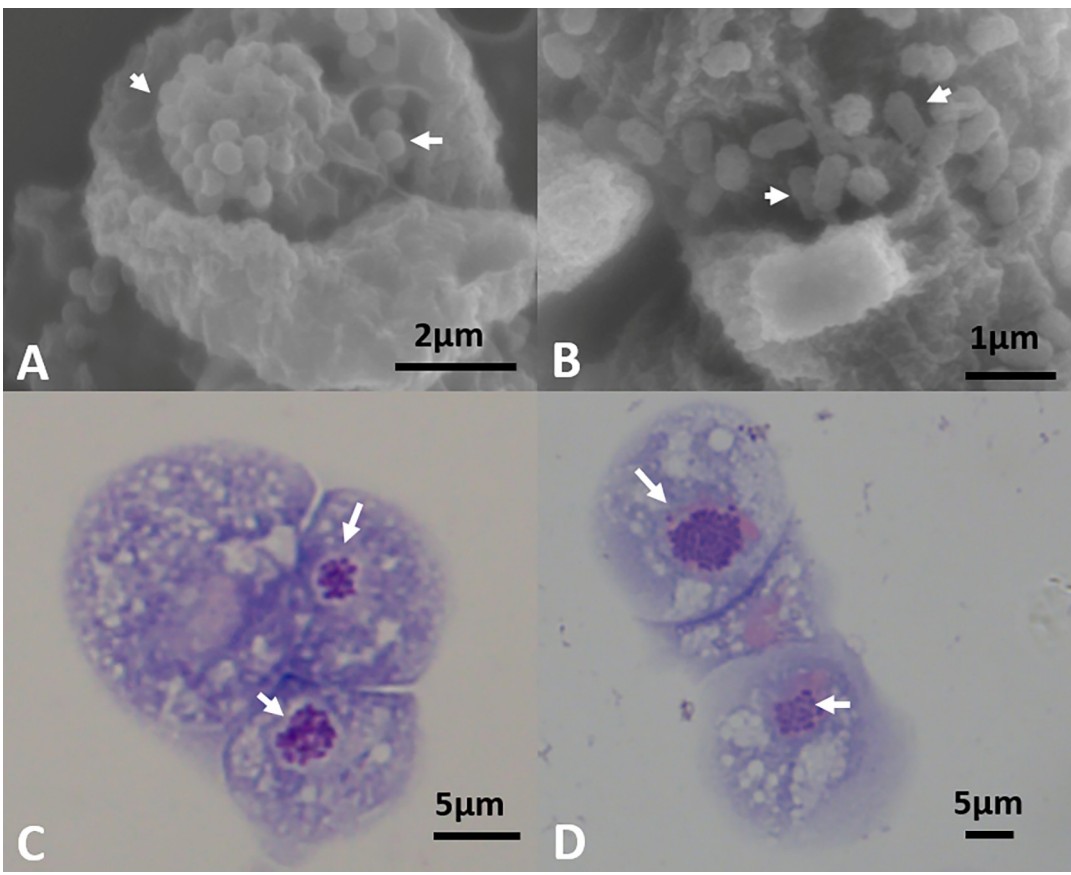

**Fig 1. Protozoa infected with "*Ca*. B. cookevillensis" (CC99) or "*Ca*. B. aquae" (HT99).** Electron micrograph (13,000x magnification) of intracellular CC99 (arrow) exhibiting coccoid morphology following exposure of the BCV by the tape ripping technique (A). Micrograph of adherent HT99 (arrow) on the surface of *A. polyphaga* exhibiting coccobacillus morphology (20,000x magnification) (B). Giemsa staining of *D. discoideum* (strain AX2) infected with CC99 (C) and HT99 (D) showing bacteria (arrow; dark purple) associated with nuclei (pink).

analyses, CC99 and HT99 have been classified as separate novel species forming distinct taxonomic lineages within the Coxiellaceae family of the order Legionellales and class Gammaproteobacteria [17]. However, recent phylogenetic analysis that included more than 100 Gammaproteobacteria genomes using concatenated amino acid alignment of 109 proteins has classified CC99 and HT99 as a separate distinct family outside Coxiellaceae [18]. Both bacteria show a close 16S rRNA similarity each other (~94%) and to the intracellular pathogens *Coxiella burnetii* (~90–91%) and *L. pneumophila* (~88%) [17], the causative agents of the zoonotic disease Q fever [19] and Legionnaires' disease [20], respectively.

Both CC99 and HT99 infect and replicate within protozoa, including *Acanthamoeba polyphaga* and *Dictyostelium discoideum* (Fig 1) [17]. CC99 can also infect and replicate in mammalian cells (S1 Fig), including phagocytic and non-phagocytic cell lines [21], also associating with or entering the nucleus. During infection of host cells, both bacteria exhibit replication within a bacteria-containing vacuole (BCV) that interacts with or enters the host cell nucleus. Within one hour after internalization, both bacteria are visible in the host cell cytosol enclosed in a vacuole. As infection progresses, the BCV traffics through the cytoplasm and eventually invades or closely associates with the nucleus of the host [21]. Within BCVs, the bacteria replicate to large numbers and eventually escape after lysis of the host cell. The mechanism of infection and intracellular replication as well as the biochemical composition of the replication vacuole of each bacterium is not yet understood, and both bacteria remain unculturable outside of host cells.

Here, we present the sequences and analyses of CC99 and HT99 genomes to gain an understanding of their genome content and insights into their biology and metabolic capacity. Information gained from the genome analyses may facilitate the understanding the mechanisms by which they invade and replicate within their host, as well as provide insight for the development of axenic media for culture outside of host cells.

## Materials and methods

### Bacterial culture

*A. polyphaga* (ATCC strain 30461), grown in 25 cm$^2$ flat-bottomed cell culture flasks in tryptic soy broth (TSB; Becton Dickinson, Franklin Lakes, NJ, USA) at 25˚C, was used to maintain and propagate both CC99 and HT99. Prior to transferring bacteria, TSB medium was removed from confluent *A. polyphaga* monolayers and cells were washed three times with sterile spring water (Carolina Biological Supply, Burlington, NC, USA) without disturbing the amoeba monolayer. Infections in amoebae were performed in sterile spring water. Co-cultures were incubated for 4–5 days at 25˚C.

### Genomic DNA extraction and purification

For whole genome sequencing, DNA was purified from host cell-free isolates following their growth in *A. polyphaga* co-culture. After complete lysis of the amoebae, bacterial cells were separated from host cells using Renografin density-gradient centrifugation as described in [22]. Briefly, amoebal debris from the bacterial lysate was removed by centrifugation at 500 *x g* for 5 minutes. The recovered supernatant was filtered through 0.8 μm PVDF filters and centrifuged at 31,000 *x g* for 30 minutes. The bacterial pellets were re-suspended in sterile phosphate buffered saline sucrose (PBSS; pH 7.4) and layered onto 30% (v/v) RenoCal-76 cushions (76% Renografin in PBSS). Tubes were filled to the top with PBSS and centrifuged at 58,000 *x g* for 30 minutes at 4˚C. The supernatant was then removed and re-suspended in cold PBSS and centrifuged at 31,000 x g for 10 minutes at 4˚C. The pellets were washed in phosphate buffered

saline (PBS) to remove any residual Renografin and purified bacteria were re-suspended in PBS.

Total DNA from host cell-free bacteria was extracted using the MasterPure™ Complete DNA Purification Kit (Lucigen, Middleton, WI, USA) following the manufacturer's instructions. Genomic DNA was quantified using a Nanodrop Spectrophotometer (Thermo Fisher Scientific, Waltham, MA, USA) and DNA degradation and contamination were examined by 1% Tris-Acetate-EDTA agarose gel electrophoresis.

### Genomic sequencing and analysis

Whole genome sequencing of purified DNA was performed using the PacBio sequel platform at Novogene Inc. (Durham, NC, USA). Raw reads generated from the PacBio sequencer were assembled using Canu v. 1.9 [23] and Falcon v. 1.8.1 [24] genome assembly programs. NCBI Prokaryotic Genome Annotation Pipeline (PGAP) [25] and RAST [26] server were used to identify protein-coding genes, rRNAs, tRNAs, and ncRNAs. Completeness of the genome was assessed using the Benchmarking Universal Single-Copy Orthologs (BUSCO) pipeline [27]. The KEGG automatic annotation server (KAAS) [28] was used to perform metabolic pathway prediction and reconstruction based on the Kyoto Encyclopedia of Genes and Genomes (KEGG) database (https://www.genome.jp/kegg/). The BioCyc [29] database was also used for metabolic pathway analyses.

The web-based programs SMART (Simple Modular Architecture Research Tool; http://smart.embl-heidelberg.de/) [30], Profile HMM database (Pfam) [31], and National Center for Biotechnology Information's Conserved Domain Database (CDD) [32] were used to identify eukaryotic or eukaryotic like domains/motifs in protein encoding genes (cut-off e-value of 0.001). Homology searches were performed against the SecReT4 [33] and EffectiveDB [34] databases. Protein encoding genes were searched against the virulence factor database (VFDB), a specialized repository of known bacterial virulence factors [35]. SignalP 4.1 (http://www.cbs.dtu.dk/services/SignalP/) and TMHMM (http://www.cbs.dtu.dk/services/TMHMM/) servers were used to identify N-terminal signal peptide sequences and transmembrane domains, respectively.

## Results and discussion

### General genome features "*Ca*. B. cookevillensis" and "*Ca*. B. aquae"

The genome of CC99 consists of 2,984,836 base pairs (bp) with an average GC content of 37.9% while the genome of HT99 consists of 3,588,707 bp with an average GC content of 39.4%. Plasmids were not experimentally identified for either bacterium nor indicated by sequence data. The protein coding densities of CC99 (89.06%) and HT99 (90.63%) were higher than the average coding density (87%) of all sequenced bacterial genomes [36], although the related pathogens *C. burnetii* and *Legionella* spp. also have high coding densities of approximately 90% [37, 38]. Genome annotation identified 2,651 protein-encoding genes (PEGs) in CC99, with approximately 52% assigned putative biological function. In contrast, annotation in HT99 identified 3,238 PEGs, with about 48% assigned putative biological function. A high proportion of PEGs were annotated as hypothetical proteins (CC99: 46.5%; HT99: 51.3%). A majority of the PEGs utilize the AUG start codon (CC99: 88.1%; HT99: 90.2%) and the remainder utilize the alternate start codons, GUG (CC99: 7.8%; HT99: 6.4%) and UUG (CC99: 4.1%; HT99: 3.4%). PEGs were distributed evenly between the forward and reverse strand (CC99: 47.94% forward, 52.06% reverse; HT99: 49.75% forward, 50.25% reverse). A total of 40 and 41 tRNA genes, representing at least one for each of the 20 amino acids, were identified in CC99 and HT99, respectively. Genes for the 3 rRNAs [5S; 23S (large subunit); 16S

**Table 1. General genome features of the "*Ca.* B. cookevillensis" and "*Ca.* B. aquae".**

| Attribute | | "*Ca.* B. cookevillensis" (CC99) | "*Ca.* B. aquae" (HT99) |
|---|---|---|---|
| Genome draft size (bp) | | 2,984,836 | 3,588,707 |
| G + C content (%) | | 37.9 | 39.4 |
| Plasmids | | NA | NA |
| Coding Density (%) | | 2,658,466 (89.06) | 3,252,581 (90.63) |
| Number of PEGs | | 2,651 | 3,238 |
| Average PEG length (bp) | | 1,002.8 | 1,004.5 |
| Median PEG length (bp) | | 815 | 824 |
| Number of rRNAs | | 3 | 3 |
| | 5s length (bp) | 120 | 119 |
| | 16s length (bp) | 1545 | 1545 |
| | 23s length (bp) | 2907 | 2904 |
| Number of tRNAs | | 40 | 41 |
| Number of sRNA | | 0 | 0 |
| Number of IS elements | | 26 | 6 |
| Average length of tRNA (bp) | | 77.6 | 77.9 |
| Hypothetical proteins (%) | | 1232 (46.5) | 1662 (51.3) |

(small subunit)] were also identified. CC99 and HT99 genome features are summarized in Table 1 and Fig 2. Complete genome annotation is available in S1 Table.

Classification of PEGs into Clusters of Orthologous Groups (COGs) [39] assigned a total of 1,910 (72.04%) and 2,177 (67.23%) PEGs to COGs in CC99 and HT99, respectively (Table 2; S1 Table). The "translation, ribosomal structure and biogenesis," "amino acid transport and

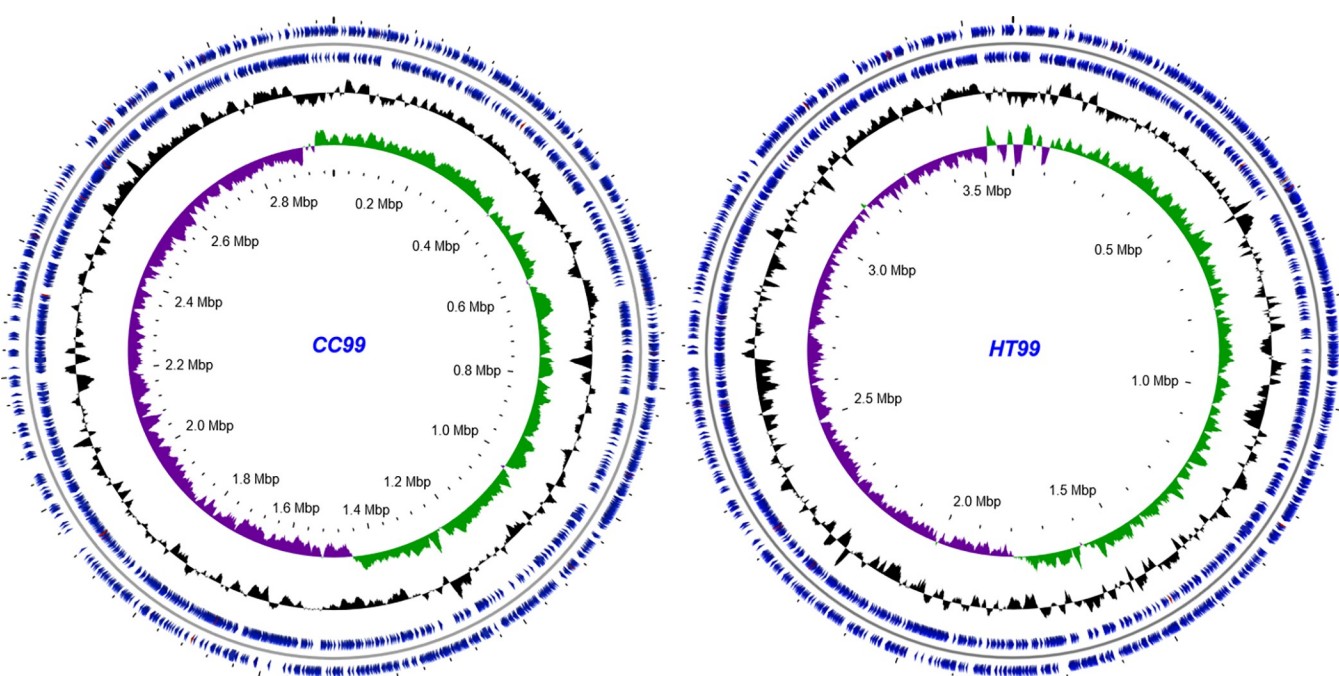

**Fig 2. Structural features of "*Ca.* B. cookevillensis" (CC99) and "*Ca.* B. aquae" (HT99) genomes.** Genome tracks show (from inner to outer) the positive (green) and negative (purple) GC skew [(C-G)/(C+G)], G+C content (black) and the coding DNA sequences (blue) located in the reverse and forward and strands. Putative origin of replication is located at the top. CGview (http://wishart.biology.ualberta.ca/cgview/) was used to construct the genome map.

**Table 2. "*Ca*. B. cookevillensis" (CC99) and "*Ca*. B. aquae" (HT99) PEGs grouped into COG functional categories.**

| | | | | | |
|---|---|---|---|---|---|
| **CELLULAR PROCESSES AND SIGNALING** | | | | | |
| | **CC99** | | **HT99** | | |
| **COG class** | **Count** | **%** | **Count** | **%** | **COG Description** |
| D | 51 | 1.92 | 52 | 1.61 | Cell cycle control, cell division, chromosome partitioning |
| M | 148 | 5.58 | 188 | 5.80 | Cell wall/membrane/envelope biogenesis |
| N | 73 | 2.75 | 75 | 2.32 | Cell motility |
| O | 84 | 3.17 | 91 | 2.81 | Post-translational modification, protein turnover, chaperones |
| T | 74 | 2.79 | 106 | 3.27 | Signal transduction mechanisms |
| U | 90 | 3.39 | 91 | 2.81 | Intracellular trafficking, secretion, and vesicular transport |
| V | 38 | 1.43 | 44 | 1.36 | Defense mechanisms |
| **INFORMATION STORAGE AND PROCESSING** | | | | | |
| A | 1 | 0.04 | 2 | 0.06 | RNA processing and modification |
| B | 1 | 0.04 | 2 | 0.06 | Chromatin structure and dynamics |
| J | 175 | 6.6 | 177 | 5.47 | Translation, ribosomal structure, and biogenesis |
| K | 103 | 3.89 | 127 | 3.92 | Transcription |
| L | 164 | 6.19 | 134 | 4.14 | Replication, recombination, and repair |
| **METABOLISM** | | | | | |
| C | 135 | 5.09 | 136 | 4.20 | Energy production and conversion |
| E | 143 | 5.39 | 148 | 4.57 | Amino acid transport and metabolism |
| F | 72 | 2.72 | 84 | 2.60 | Nucleotide transport and metabolism |
| G | 64 | 2.41 | 82 | 2.53 | Carbohydrate transport and metabolism |
| H | 107 | 4.03 | 101 | 3.12 | Coenzyme transport and metabolism |
| I | 71 | 2.68 | 91 | 3.37 | Lipid transport and metabolism |
| P | 90 | 3.39 | 109 | 4.68 | Inorganic ion transport and metabolism |
| Q | 51 | 1.92 | 67 | 2.07 | Secondary metabolites biosynthesis, transport, and catabolism |
| **POORLY CHARACTERIZED** | | | | | |
| S | 328 | 12.4 | 454 | 14.02 | Function unknown |

metabolism," and "cell wall/membrane/envelope biogenesis" classes (COG classes J, E, M) were the most prominently represented categories in both bacteria (Table 2).

As expected, genes for the necessary components of the genetic information processing machinery were present in both CC99 and HT99. The macromolecular synthesis (MMS) operon, containing the genes *rpsU*, *dnaG*, *rpoD*, which encode essential products for the initiation of protein, DNA and RNA synthesis [40], was identified in both bacteria. Genes encoding for proteins involved in DNA replication and transcription, including ATP-independent DNA topoisomerase I, ATP-dependent topoisomerase IV (subunits A and B), DNA gyrase (subunits A and B), RNA polymerase core subunits (*rpoA*, *rpoB*, *rpoC*, *rpoZ*), RNA polymerase sigma factors (*rpoD*, *rpoS*, *rpoE*, *rpoH*, *rpoN*), transcription termination factor (*rho*), RNA polymerase associated proteins (*nusA*, *nusB*), transcriptional anti-terminator (*nusG*), stringent starvation proteins (*sspA*, *sspB*), transcription elongation factors (*greA*, *dksA*) and transcription-repair coupling factor were identified in both bacteria. Genes for the large and small ribosomal subunit proteins, essential components of ribosomal structures, were also identified in both bacteria.

Genes dedicated to DNA repair and recombination processes, including genes for DNA repair (*adaA*, *adaB*, *radA*, *dam*, *mutT*, *pcrA*, *recD*, *recF*, *recO*, *recR*, *recN*, *recG*, *recA*, *recX*, *rmuC*, *ssb*, *xseA*, *xseB*,), excision repair (*uvrABCD*), DNA mismatch repair (*mutS*, *mutL*), base excision repair (*recJ*, *mutM*, *mutY*, *tag*, *ung*, *alkA*), and recombinational repair (*ruvA*, *ruvB*,

*ruvC*, *recJ*) were identified in both bacteria. The presence of a large number of DNA recombination and repair genes suggests that these bacteria have high recombination capabilities. Genes encoding proteins involved in protein folding, including chaperones (*groEL*, *groES*, *hrcA*, *grpE*, *dnaK*, *dnaJ*, *hscA*, *hscB*, *htpG*, *clpA*, *clpB*) and disulfide interchange proteins (*dsbA*, *dsbB*, *dsbC*, *dsbD*, *dsbE*) were also identified in both bacteria.

## Metabolism

Due to their obligate nature, very little is known about the metabolic capabilities of CC99 and HT99. Attempts to grow them in host cell-free media have, thus far, been unsuccessful. Obligate reliance on their host for growth has limited phenotypic and genetic experimental studies of these bacteria. Analyses of bacterial genomes have made it possible to gain insights into their metabolic capacity.

**Carbohydrate metabolism.** Except for glucokinase (*glk*), we identified all the genes encoding enzymes required to generate pyruvate through the Embden-Meyerhof-Parnas (EMP) pathway in both CC99 and HT99 (Fig 3). *C. burnetii* likewise is missing the gene for glucokinase, nor does it require glucose for replication; however, glycolytic activity has been reported for the bacterium such that glucose use increases biomass in axenic culture with amino acid supplementation [41, 42]. *L. pneumophila* has a complete glycolytic pathway (Fig 3). Phosphoenolpyruvate (PEP)-dependent phosphotransferase systems (PTS), generally used for carbohydrate uptake in bacteria, were not identified in either bacterium. A glycerol uptake system or a glycerol facilitator gene (*glpF*) was also absent. In CC99, genes encoding for glycerol kinase (*glpK*) and an aerobic glycerol-3-phosphate dehydrogenase (*glpD*) were identified, which suggests that CC99 may be capable of phosphorylating glycerol (obtained by passive diffusion or through degradation of glycerol-containing lipids from the host by lipases) to glycerol-3-phosphate, which could then be converted to dihydroxyacetone phosphate, an intermediate of glycolysis that could shuttled through the glycolytic pathway to generate pyruvate [43].

The Entner-Doudoroff (ED) pathway appears to be completely absent in both CC99 and HT99. The ED pathway is also absent in *C. burnetii*, but it is the predominant glycolytic pathway for *L. pneumophila* [44, 45]. Genes for enzymes involved in the non-oxidative branch of the pentose phosphate pathway (PPP) were identified (Fig 3), indicating a capability for producing pentose phosphates (required for nucleic acid synthesis) and erythrose phosphates (precursors for aromatic amino acids) in both bacteria. However, key genes of the oxidative branch of the PPP, including glucose 6-phophate dehydrogenase (*zwf*), gluconolactonase (*pgl*), and 6-phosphogluconate dehydrogenase (*gnd*), were absent in both bacteria (Fig 3), suggesting that they may not be able to produce NADPH utilizing the PPP. Other intracellular bacteria, including *C. burnetti*, *L. pneumophila*, and *Francisella tularensis* are also missing genes for a complete oxidative branch of the PPP [38, 46–48]. Both CC99 and HT99 are missing genes encoding fructose 1,6-bisphosphatase (*fbp*) and PEP carboxykinase (*pckA*), key enzymes of gluconeogenesis that generate intermediate precursors for other biosynthetic pathways (Fig 3). Genes for the pyruvate dehydrogenase (*pdh*) complex, as well as all the genes tricarboxylic acid (TCA) cycle were identified in both bacteria, suggesting capability for ATP production via substrate-level phosphorylation (Fig 3). We identified genes for PEP synthase (*pps*) and NAD- and NADP-dependent malic enzymes (*mea*) in both bacteria. Via these enzymes, they may be able to convert malate to pyruvate and generate PEP [49]. Neither bacterium may be capable of generating carbohydrates from acetyl-CoA subunits as genes for enzymes of the glyoxylate shunt, isocitrate lyase (*aceA*) and malate synthase (*glcB*), were not identified. Although the glyoxylate pathway is common in aerobic bacteria and has been associated with virulence, it is

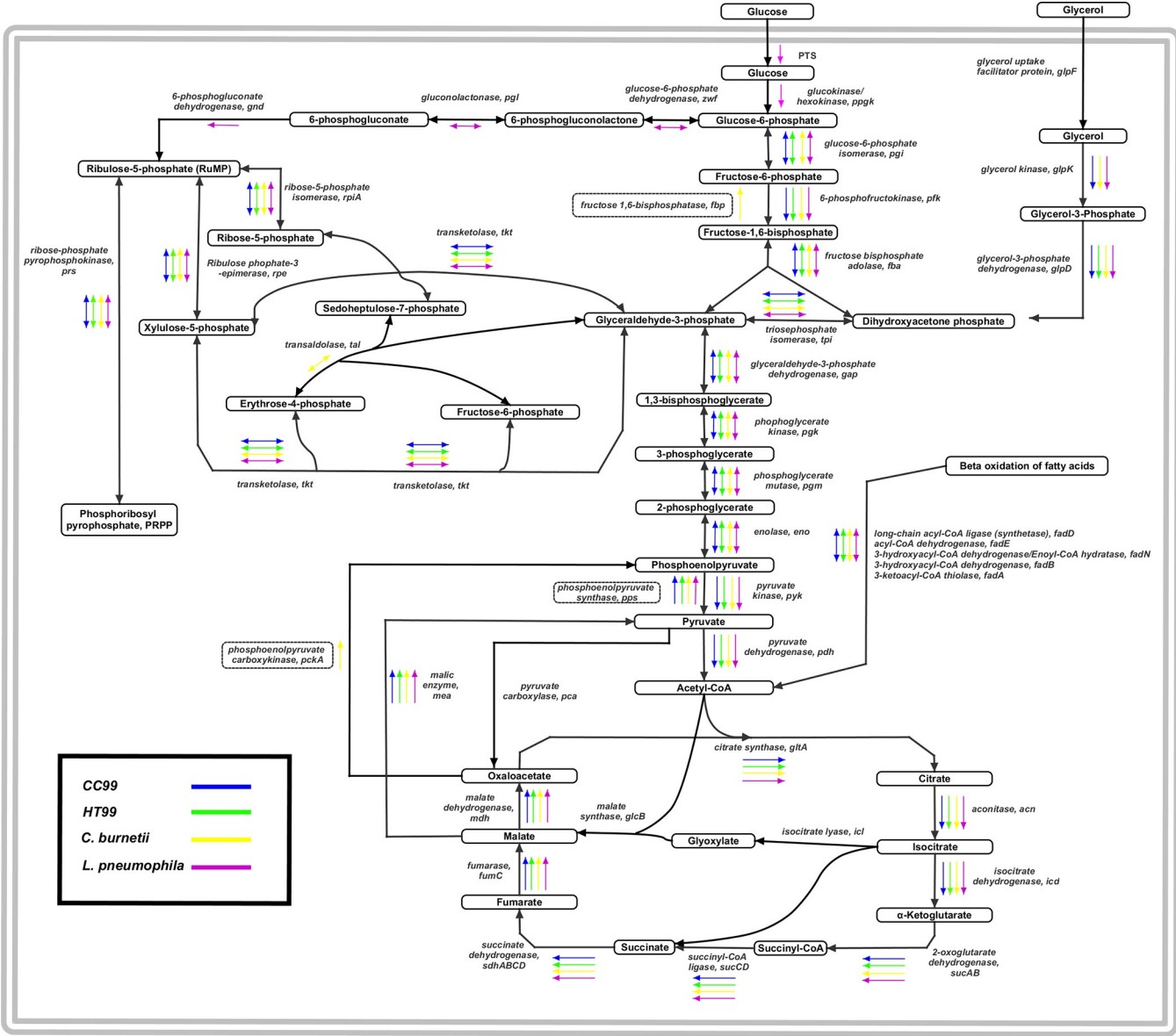

**Fig 3. Genome inferred central carbohydrate metabolic pathway of "*Ca.* B. cookevillensis" (CC99) and "*Ca.* B. aquae" (HT99) with comparisons to *C. burnetii* and *L. pneumophila*.** Genes encoding enzymes involved in the EMP of glycolysis (except for glucokinase) and non-oxidative branch of the PPP were identified in both CC99 and HT99 (indicated by blue and green arrows, respectively). Genes for enzymes involved in the oxidative branch of PPP pathway and the ED pathway were absent. Genes encoding enzymes involved in the TCA cycle were present in both bacteria. Genes encoding enzymes for these pathways in *C. burnetii* (yellow arrows) and *L. pneumophila* (pink arrows) are included for comparison.

incomplete in many intracellular bacteria including *C. burnetti*, *L. pneumophila*, *F. tularensis*, *Listeria monocytogenes*, and *Rickettsia* spp. [38, 50, 51].

Genes encoding enzymes involved in aerobic respiration were present in both bacteria, including 14 subunits of respiratory NADH:ubiquinone oxidoreductase (NADH dehydrogenase complex; complex I), 4 subunits of respiratory succinate:quinone oxidoreductase (succinate dehydrogenase complex; complex II), and 3 subunits of respiratory ubiquinone-cytochrome c (cytochrome c reductase; complex III). Genes for a terminal cytochrome oxidase complex, including cytochrome bd (*cydAB*) and cytochrome c (*coxCBA*, *cyoE*) were also

identified. An additional Cbb3-type cytochrome oxidase was also identified in CC99. The cytochrome bd oxidase, shown to have an increased affinity for oxygen [52], may allow CC99 to survive under limiting oxygen levels. Genes encoding the F-type ATPase ($F_1F_0$ ATP synthase) units that catalyze ATP hydrolysis were also present in both bacteria. Taken together, the presence of these genes suggests that both bacteria likely rely on proton motive force-driven aerobic respiration for ATP production.

**Lipid metabolism.**   In both CC99 and HT99, genes for enzymes involved in the initiation step (*accADCB*, *acpP*, *acpS*, *fabD*, *fabH*) and chain elongation cycle (*fabG*, *fabZ*, *fabI*, *fabF*) of the type II fatty acid synthesis pathway were identified, suggesting the capability of producing fatty acids. A gene homologue of *fabV* encoding an enoyl-acyl carrier protein reductase, implicated in resistance to the antibacterial triclosan in *Vibrio cholera* and *Pseudomonas aeruginosa* [53, 54], was identified in HT99. Genes for 3-hydroxydecanoyl dehydratase (*fabA*) and 3-oxoacyl synthase 1 (*fabB*), acyl-carrier protein homologs which together control the level of unsaturated fatty acid synthesis in *Escherichia coli* [55], were not identified. Absence of these genes suggests that both CC99 and HT99 may not be capable of synthesizing unsaturated fatty acids. Indeed, lipid profile analyses have previously shown that both CC99 and HT99 contain straight chain fatty acids [17].

Genes encoding an outer membrane protein (*fadL*) and a long-chain acyl-CoA synthetase (*fadD*) were identified in both bacteria. In *E. coli*, these proteins are responsible for transporting fatty acids via a transport/acyl-activation mechanism [55] and may have similar functions in CC99 and HT99. Genes encoding the necessary enzymes to generate ATP from β-oxidation of fatty acids (*fadE*, *fadB*, *fadA*) were also identified in both bacteria (Fig 3). Via β-oxidation pathway, both CC99 and HT99 may be able to degrade long chain fatty acids into acetyl-CoA, which can then be further oxidized via the TCA cycle for ATP production.

Both bacteria lack genes involved in the mevalonate pathway for isoprenoid biosynthesis but have genes for enzymes of the non-mevalonate pathway for isoprenoid biosynthesis. The non-mevalonate pathway, common in most gram-negative bacteria, generates the five-carbon isoprenoid precursors isopentyl diphosphate and dimethylallyl diphosphate which can be modified to make diverse organic molecules involved in important biological functions in the cell, including electron transport and peptidoglycan biosynthesis [56]. Conversely, *C. burnetii* and *L. pneumophila* have genes for the mevalonate but not the non-mevalonate pathway [38, 57]. The mevalonate pathway has been associated with predatory bacterial species as well as the intracellular bacteria, *Teredinibacter turnerae* and 'Ca. Liberibacter asiaticus' [57]. Both bacteria encode enzymes for synthesizing phosphatidylethanolamine (PE) and phosphatidylserine, important phospholipid membrane components. However, they are missing the genes encoding phosphatidylglycerophosphatase (*pgpA*, *pgpB*, *pgpC*) and cardiolipin synthetase (*cls*), key enzymes involved in phosphatidylglycerol and cardiolipin biosynthesis, although these genes are found in *C. burnetti* and *Legionella* spp. [58, 59].

**Nucleotides, amino acids and cofactor metabolism.**   Both CC99 and HT99 have genes encoding proteins involved in the synthesis of purines and pyrimidines *de novo*. Genes for key enzymes involved in the initial synthesis of uridine monophosphate (UMP) from phosphoribosyl pyrophosphate (PRPP) and L-glutamine (*carA*, *carB*, *pyrB*, *pyrC*, *pyrD*, *pyrE*, *pyrF*) and for converting UMP to uridine diphosphate/triphosphate (UDP/UTP) and cytidine diphosphate/triphosphate (CDP/CTP) (*cmk*, *ndk*, *pyrG*, *pyrH*) were identified. Genes for enzymes involved in generating inosinic acid (IMP) (a purine precursor) from L-glutamine and ribose-5-phosphate (*prs*, *purF*, *purD*, *purN*, *purL*, *purM*, *purK*, *purE*, *purC*, *purH*) and for the subsequent conversion of IMP to adenosine monophosphate (AMP) (*purA*, *purB*) and guanosine monophosphate (*guaAB*) were also identified.

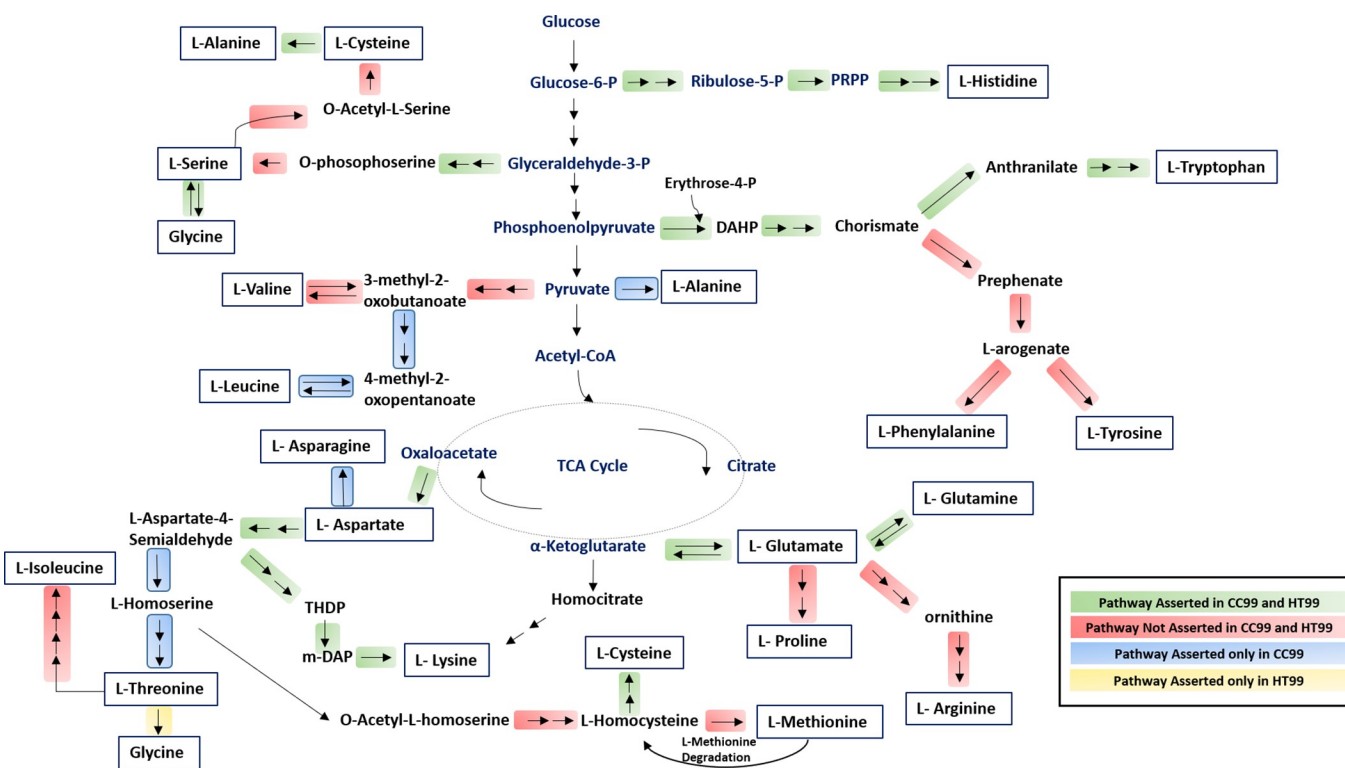

**Fig 4. Amino acid biosynthesis pathways of "*Ca.* B. cookevillensis" (CC99) and "*Ca.* B. aquae" (HT99).** Genome inferred amino acid pathways appear to be reduced in both bacteria. Only 11 of the 20 amino acid pathways could be asserted for CC99 while only 8 could be asserted for HT99.

Amino acid biosynthesis pathways appear to be reduced in both CC99 and HT99 (Fig 4, S1 Text, S2 Table). For CC99, 11 (Ala, Asn, Asp, Glu, Gln, Gly, His, Lys, Ser, Thr and Trp) of the 20 amino acid pathways could be asserted while only 8 (Asp, Glu, Gln, Gly, His, Lys, Ser, and Trp) could be asserted for HT99 (Fig 4). A high degree of amino acid auxotrophy suggests that both bacteria likely rely on their hosts for obtaining the necessary amino acids and intermediates. Both *C. burnetti* and *L. pneumophila* are also auxotrophic for several amino acids yet have potential mechanisms to scavenge amino acids from their host cells. *C. burnetti* has been reported to up-regulate autophagy resulting in the release of free amino acids, and *L. pneumophila* induces proteasome degradation of host cell proteins to increase in intracellular amino acids [60, 61]. *C. burnetti* and *L. pneumophila* have also both been grown axenically in media containing only amino acids [41, 62] which suggests that amino acid-based medium may be developed for the axenic growth of bacteria CC99 and HT99.

Several genes involved in cofactor biosynthesis pathways were identified. Genes for enzymes required to catalyze the biosynthesis of riboflavin [precursor for flavin mononucleotide (FMN) and flavin adenine dinucleotide (FAD)] from guanosine triphosphate and ribulose-5-phosphate were identified in both CC99 and HT99. A gene encoding a bifunctional riboflavin kinase/FMN adenylyltransferase (*ribF*), involved in phosphorylation of riboflavin to the FMN and subsequent adenylation of FMN to FAD, was also present in both bacteria. FMN and FAD are important cofactors for many metabolic enzymes that involve oxidation-reduction reactions. Genes for *de novo* synthesis of NAD from L-aspartate or L-tryptophan were not identified. However, genes encoding NAD salvage-specific enzymes that synthesize NAD from nicotinamide or nicotinic acid, including nicotinate phosphoribosyltransferase (*pncB*), nicotinamidase (*pncA*), NAD synthetase (*nadE*) and nicotinate-nucleotide adenylyltransferase

(*nadD*), were identified in both bacteria. Both bacteria also encode NAD kinase (*nadK*), a key enzyme which catalyzes the phosphorylation NAD to from NADP [63].

Biotin, derived from pimelic acid, is an important cofactor for many carboxylation, decarboxylation and transcarboxylation reactions [64]. Genes (*bioF*, *bioA*, *bioD*, *bioB*) encoding enzymes involved in a four-step path conversion of pimeloyl-CoA to biotin (second stage of biotin biosynthetic pathway) [65] were identified in both bacteria. However, genes involved in the synthesis of a pimelate moiety (first stage of biotin biosynthetic pathway), including genes encoding pimeloyl-CoA synthetase *(bioW)* and an enzyme of the cytochrome P450 family (*bioI*) involved in direct conversion of pimelic acid to pimeloyl-CoA [66] are missing in both bacteria. Both bacteria are also missing the gene encoding pimeloyl methyl ester esterase (*bioH*). In *E. coli*, this enzyme acts synergistically with malonyl-O-methyltransferase (encoded by *bioC*) to produce a pimelate moiety that serves as a carbon backbone in the early steps of biotin biosynthesis [67]. Absence of these genes suggests that both bacteria may synthesize the pimelate moiety precursor via an alternate pathway, or they may not be able to use pimelic acid as a source for biotin synthesis. Genes encoding transmembrane component (*bioN*) and a substrate-specific component (*bioY*) for a biotin transporter protein are absent in both bacteria and so they may not be capable of transporting biotin.

Genes encoding the necessary enzymes required to convert pantothenate (vitamin B5) to coenzyme A, an essential cofactor in many important metabolic processes, were identified in both bacteria. However, they are missing essential genes for enzymes involved in the *de novo* synthesis of pantothenate from aspartate and α-ketoisovalerate (*panD*, *panE*), suggesting that they likely import pantothenate precursors from host cells. Genes for the biosynthesis of thiamine, including genes encoding enzymes involved in the biosynthesis of thiazole and pyrimidine moieties (thiamine precursors) were identified in both bacteria. CC99 encodes a complete gene set for the folate biosynthesis pathway. HT99, however, is missing important genes in this pathway, including aminodeoxychorismate lyase (*pabAc*).

## Motility, secretion and adhesion

The bacterial flagellum, assembled from as many as 40 different protein components, is a complex motility organelle composed of a basal body, a curved flexible hook, and a long helical propagating filament [68]. In both CC99 and HT99, genes encoding the flagella hook/filament (*flgE*, *flgL*, *flgK*, *fliC*, *fliD*), motor/switch (*motA*, *motB*, *fliM*, *fliN*, *fliG*), basal body (*flgG*, *flgH*, *flgJ*, *flgI*, *flgF*, *flgC*, *flgB*, *fliE*, *fliF*), and flagella export apparatus (*flhA*, *flhB*, *fliP*, *fliQ*, *fliR*, *fliI*) components were identified (Fig 5). Most of these genes are located within a single locus. Both CC99 and HT99 have been shown to be motile by a single, polar flagellum [17].

Homologs of *flhD* and *flhC* genes, which regulate the transcriptional activation of class II flagellar genes in *E. coli* [69], were not identified. However, gene homologs encoding sigma factor (σ) 54 (*rpoN*), transcriptional regulator FleQ (*fleQ*) and trascription inititation factor FilA (*fliA*) were identified in both bacteria. During the transmissive life-cycle of *L. pneumophila*, the regulatory protein FleQ together with σ54, enhance the expression of flagellar class II genes which encode protein components involved in the assembly of flagellar basal body, flagellar hook and regulatory proteins while FilA induces the expression of flagellar class III genes (encoding motor/switch proteins) and class IV (encoding filament and filament cap proteins) genes [70, 71]. The presence of these transcription regulator homolog genes suggest flagellar gene expression in CC99 and HT99 is similar to that of *L. pneumophila*.

Genes encoding proteins that recognize and bind signal peptides of the twin-arginine translocase (TAT)-dependent folded substrates (*tatB*, *tatC*) were missing, while genes for the general secretion (Sec) dependent translocon which transports unfolded proteins across the

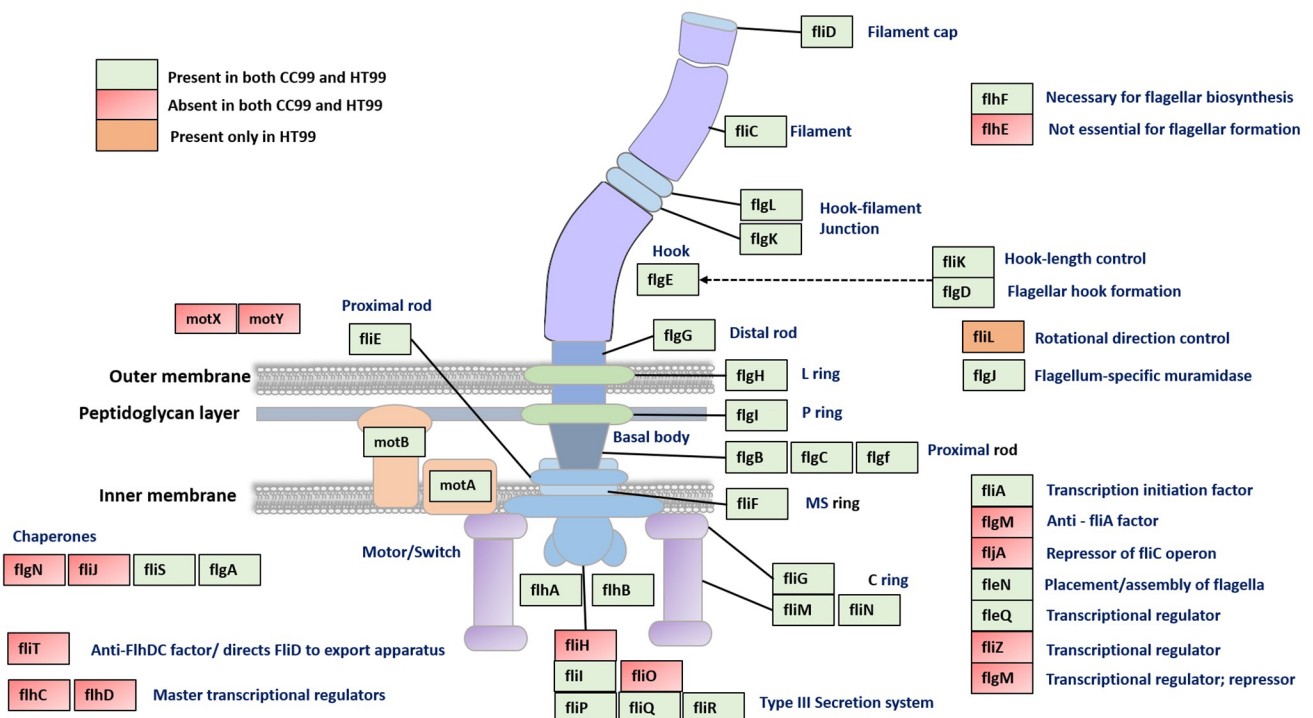

**Fig 5. Genes involved in the synthesis of flagella in bacteria.** Gene homologs encoding flagellar proteins identified in "*Ca*. B. cookevillensis" (CC99) and "*Ca*. B. aquae" (HT99). Genes encoding motor/switch, basal body, hook, filament, filament cap proteins were identified.

cytoplasmic membrane were present in both bacteria. A set of genes encoding T1SS components were identified in both bacteria, including genes encoding an ABC transporter, an outer membrane-spanning porin TolC, and a periplasmic membrane fusion protein of the HlyD family. Type I secretion systems (T1SS), which belong to a family of ATP-binding cassette (ABC) transporters, secrete unfolded substrates in a one-step process directly from the cytoplasm to extracellular milieu without a periplasmic intermediate [72]. T1SSs secrete a diverse range of substrates involved in bacterial pathogenesis, including proteases, bacteriocins and adhesins [73]. Moreover, genes encoding T1SS-secreted agglutinin repeat-in-toxins (RTX), which are diverse multifunctional proteins with important roles in pathogenesis in many bacterial species, including *V. cholerae* [74] and *L. pneumophila* [75], were also identified in both bacteria. The presence of T1SS genes and genes encoding T1SS-secreted toxins suggests that CC99 and HT99 each have a functional T1SS.

Genes for components of functional Type II, III, V or VI secretion systems were not identified in either CC99 or HT99. However, genes for the Dot/Icm (Defect in organelle trafficking/Intracellular multiplication) type IV secretion system (T4SS) were identified in both bacteria. The dot/icm T4SS is a class B type IV secretion apparatus composed of at least 27 protein components, that spans the inner and outer bacterial cell membrane [76]. Found in pathogens such as *L. pneumophila* and *C. burnetii*, this specialized transport apparatus allows the delivery of a large repertoire of virulence factors (dot/icm substrates) into host cells [38, 76, 77]. Within host cells, dot/icm substrates have been shown to manipulate various host cell processes allowing the establishment of replication vacuole and intracellular growth in *L. pneumophila* and *C. burnetii* [78–80]. We identified 17 and 18 of the 27 *L. pneumophila* dot/icm genes in CC99 and HT99, respectively (Fig 6). Dot/icm genes in CC99 and HT99 share significant homology to

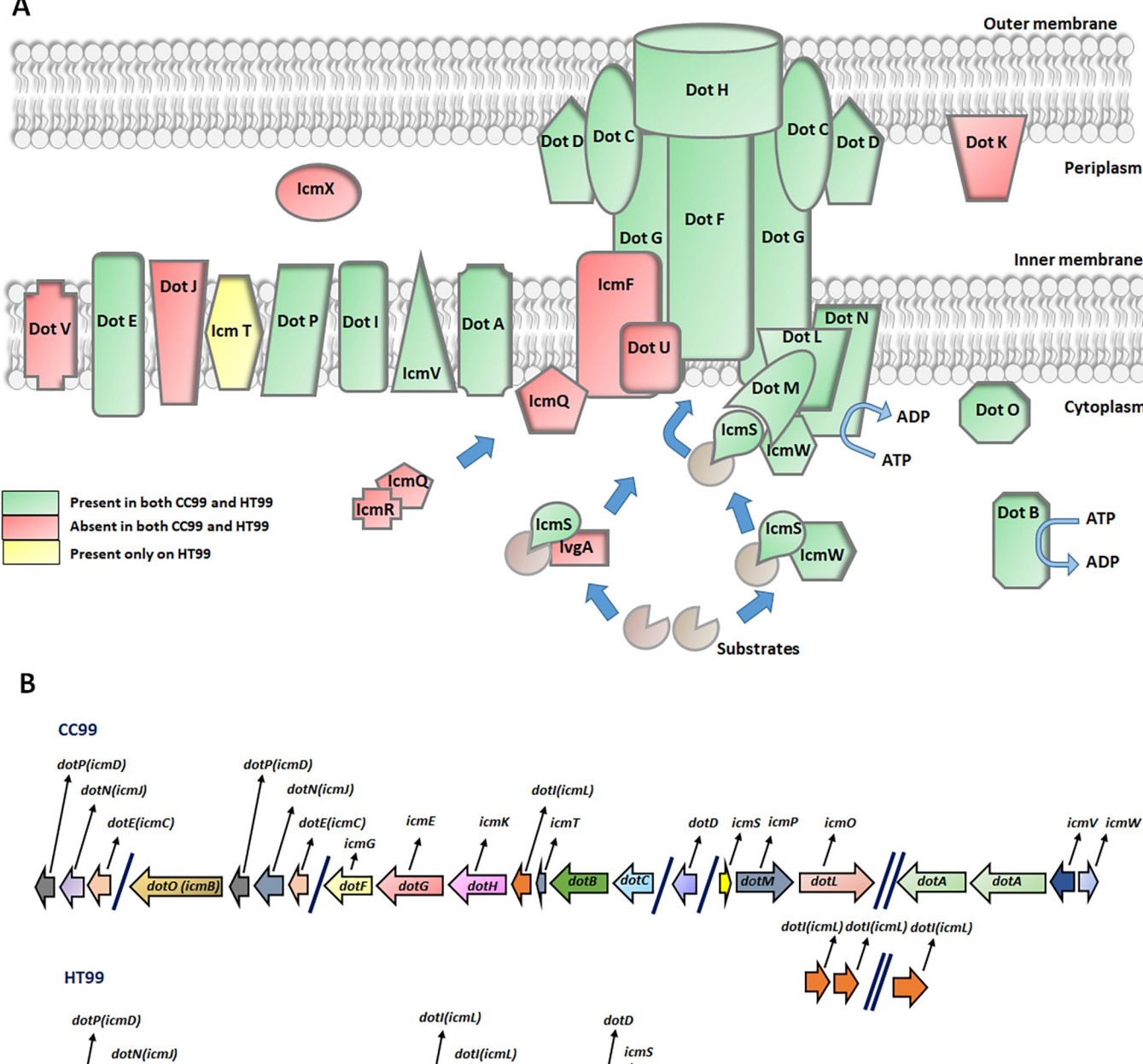

**Fig 6. Dot/icm T4SS in "*Ca*. B. cookevillensis" (CC99) and "*Ca*. B. aquae" (HT99) with comparisons to *C. burnetii* and *L. pneumophila*.** (A) Presumed location and topological relationships of dot/icm proteins. Figure adapted from [77]. (B) Genomic organization of the genes encoding dot/icm proteins in CC99 and HT99. Genes *dotC*, *dotD*, *dotF/icmG*, *dotG/icmE* and *dotH/icmK* encode the core proteins of the complex. Genes *icmS* and *icmW* encode components involved substrate recognition and secretion. Both bacteria are missing the gene encoding LvgA, a protein identified as a potential fifth chaperone to IcmS. Genes *dotA*, *dotE*, *dotI*, *dotP* and *icmV* which encode the inner membrane proteins are also present in both bacteria. Genes *dotI* and *dotJ* (absent in both bacteria) encode for an integral inner membrane protein. Genes *dotL*, *dotM* and *dotN* encode membrane proteins components involved in recruitment of effector proteins. Genes *dotB* and *dotO* encode cytoplasmic ATPase complex proteins. Genes *dotV* and *dotJ* (encoding inner membrane proteins); *icmQ* (encoding a pore-forming cytoplasmic protein) and *icmR* (encoding a chaperone for IcmQ) are absent in both bacteria.

those of *L. pneumophila* (S3 Table). The presence of dot/icm genes suggests that both bacteria likely utilize the dot/icm T4SS to deliver effectors to host cells to mediate successful infection and intracellular replication.

Several type IV pili (Tfp) genes were identified in both bacteria. Evolutionarily related to components of type II secretion systems, Tfp are multifunctional appendages that are associated with virulence functions, such as promoting host cell adherence, twitching motility, and biofilm formation in many bacteria species including, *Legionella* species [81] and *F. tularensis* [82]. Genes for major prepilin (*pilA*), assembly proteins (*pilB*, *pilC*, *pilD*, *pilN*, *pilQ*), minor pilins (*pilE*, *pilV*, *pilW*, *pilX* and *fimT*), ATPase component (*pilU*), twitching motility protein (*pilT*) and a prepilin peptidase/N-methyltransferase (*pilD)* were identified in both CC99 and HT99. *C. burnetii* also contains a number of genes for Tfp assembly but is missing pilT and pilU genes that are critical for pilus function [83]. A functional Tfp expressed on the surface of CC99 and HT99 may play a role in promoting adherence of these bacteria to host cells.

## Effector proteins

Many intracellular bacteria rely on effectors to successfully promote their uptake, subvert essential cellular pathways, acquire essential nutrients, and suppress host immune responses [84]. When translocated to host cells, effectors can exert their function either by activating/inhibiting the function of host proteins or by mimicking the function of endogenous host proteins [85]. Using bioinformatic guided approaches, we identified 218 and 286 genes that encode putative effectors in CC99 and HT99, respectively (S4 Table). Among those identified were genes encoding proteins with domains and/or motifs frequently or primarily found in eukaryotic proteins. Proteins containing eukaryotic or eukaryotic-like domains and/or motifs, have been described in many intracellular bacteria (thought likely to have been acquired via interdomain horizontal gene transfer) such as *L. pneumophilia* and *C. burnetti* [86, 87].

In CC99 and HT99, we identified 26 and 63 genes encoding proteins containing eukaryotic-like ankyrin repeats (ANK), respectively (S4 Table). ANKs are protein-protein interaction motifs of tandemly repeated modules of 30–34 amino acids which are present in eukaryotic proteins involved in various cellular functions, including transcription regulation, signal transduction, vesicular trafficking, and cytoskeleton integrity [88]. Proteins containing ANKs have been identified in intracellular pathogens such as *L. pneumophilia* [89], *C. burnetti* [90], *Orientia tsutsugamushi* [91] and *Wolbachia* spp. [92]. Many of these proteins have effector functions that promote host-cell invasion and replication by interfering with host cell functions such as vesicular transport and preventing pathogen-induced apoptosis [93].

We also identified genes encoding proteins with tetratricopeptide repeats (TPR) (CC99: 13; HT99: 7) and Sel1 repeats (CC99: 6; HT99: 8) in both bacteria (S4 Table). TPRs, typically consisting of 34 amino acid residues arranged in tandem repeats, also facilitate protein-protein interactions and assembly of multi-protein complexes [94]. Proteins containing TPR motifs are also present in both *C. burnetii* [86] and *L. pneumophilia* [95]. TPR motifs have also been implicated in virulence-associated functions of other bacterial pathogens such as *O. tsutsugamushi* [96] and *P. aeruginosa* [97]. The Sel1 repeat (SLR) motif, a subclass of the TPR motif with a similar consensus sequence but a variable length, has also been implicated in virulence of bacterial pathogens [98, 99].

Genes encoding proteins with eukaryotic-like serine/threonine protein kinases (STPKs; CC99: 18; HT99: 23), leucine rich repeats (LRR; CC99: 6; HT99: 6), F-Box (CC99: 1; HT99: 6), and U-Box (CC99: 0; HT99: 1) domains were also identified (S4 Table). In eukaryotic organisms, STPKs control phosphorylation states of substrates involved in intracellular signaling pathways. By adding or removing phosphate groups of protein substrates, STPKs function as on/off switches to activate or deactivate specific intracellular signaling pathways [100].

Bacterial effectors with structural and functional similarities to eukaryotic STPKs and phosphatases have been identified in several intracellular pathogens, including *L. pneumophilia* [101], *C. burnetti* [86], and *M. tuberculosis* [102]. LRR-containing proteins have been implicated in the virulence of intracellular pathogens such as *Listeria monocytogenes* [103] and *Salmonella enterica* [104]. F-box and U-box domains have also been identified in genomes of numerous human and plant bacterial pathogens [105].

## Stress genes

During colonization, infection, and transmission from the host, intracellular bacteria can encounter many stresses, such as limited nutrients, changes in pH, temperature and osmolarity, and exposure to oxidative stress and toxic molecules, such reactive oxygen species (ROS), peroxides, metals, and antibiotics. Therefore, bacteria must be able to quickly respond and adapt to these stresses. In both CC99 and HT99, we identified several genes involved in oxidative stress responses, including catalase (*katE*), superoxide dismutase [Fe] (*sodB*), superoxide dismutase [Cu-Zn] (*sodC*), cytochrome c551 peroxidase (*ccpA*), glutathione reductase (*gor*), thioredoxin peroxidase (*btuE*), alkyl hydroperoxide reductase C (*ahpC*), and rubredoxin (*rubA*). We also identified a gene encoding the catalase peroxidase enzyme (*katG*) in CC99. This enzyme has been shown to play a role in replication of *Mycobacterium tuberculosis* and *F. tularensis* in host cells by contributing to resistance of ROS and reactive nitrogen species (RNS) mediated killing [106, 107]. Other stress response genes identified include stringent starvation protein genes (*sspA*, *sspB*), *hfl* operon genes (*hflq*, *hflK hflC*), and periplasmic stress sensor genes (*degS*, *rseA*).

Both bacteria contain the gene (*relA/spoT*) encoding a putative RelA/bifunctional synthetase/hydrolase SpoT homolog protein. Also present in *L. pneumophilia* and *C. burnetti* [108], RelA/SpoT protein controls the synthesis and the degradation of the small alarmone nucleotides, guanosine tetraphosphate (ppGpp) and guanosine pentaphosphate (pppGpp). These nucleotides are collectively referred to as (p)ppGpp and are synthesized in response to high stress and low nutrient conditions, allowing gram-negative bacteria to initiate an adaptive global physiological response known as a stringent response [109]. Accumulation of (p)ppGpp in bacterial cells (in cooperation with the small RNA polymerase binding protein DksA) triggers a response resulting in modification of RNA polymerase and control of transcriptional activity such that bacteria rapidly express factors crucial for nutrient acquisition and stress survival [109]. Besides promoting physiological adaptation during nutritional starvation, this stringent response mediated by (p)ppGpp is involved in pathogenesis of several pathogens, including promoting adherence of enterohemorrhagic (EHEC) *E. coli* [110], promoting invasion of host cells in *S. enterica* serovar Typhimurium [111], and transmission of *L. pneumophila* into host cells [112].

We also identified genes encoding putative two-component systems (TCS) in both bacteria. Widely found in bacteria, TCS are signal transduction pathways, composed of a sensor histidine kinase (HK) and a response regulator (RR) protein, typically encoded by a pair of adjacent genes, that allow bacteria to detect and mediate a response to changes in the environment [113]. Among those identified, include *fleS-fleR* which regulate expression of flagellar genes in *P. aeruginosa*, [114], *cheA-cheY* which regulate bacterial chemotaxis through counterclockwise/clockwise rotation in *Bacillus subtilis* [115], *barA-uvrY* which regulate efficient switching between different carbon sources in *E. coli* [116], and *pilS-pilR* which regulate fimbriae expression in *P. aeruginosa* [117].

## Conclusion

Here, we present genomic descriptions and analyses of two intracellular bacteria isolated from amoebae found in human-constructed water systems. Both 'Ca. B. cookevillensis' (strain

CC99) and 'Ca. B. aquae' (strain HT99) have been described as bacterial obligate intracellular parasites of amoebae (recently coined as BOIP by Sanchez and Omsland, 2021 [118]). CC99 is also an obligate intracellular parasite of mammalian cell lines. Both bacteria replicate within BCVs that are closely associated with or within the nucleus, resulting in lysis of the host cell [21]. Although, this genomic study provides insight into gene products that contribute to the metabolism and virulence of these bacteria, it still remains unclear why CC99 but not HT99 is able to infect mammalian cells. Notable differences between these two bacteria include their amino acid auxotrophies (9 for CC99 and 12 for HT99) and the lack of a complete folate biosynthesis pathway in HT99. Ongoing efforts in our laboratories investigating intracellular trafficking and the compartments with which these bacteria associate, as well as RNA sequencing analyses during different stages of infection will help to elucidate such differences. The obligate lifestyles of these bacteria have also limited genetic studies or manipulations of these bacteria; however, this study has identified significant nutritional deficiencies for amino acids that provide insight into development of media for axenic growth.

## Supporting information

**S1 Fig. Electron micrograph of cells infected with "*Ca*. B. cookevillensis" (CC99).** Transmission electron micrograph of Thp1 cell infected with CC99. CC99-CV appears within the nucleus at 24-hour post infection (12,000x magnification).
(TIF)

**S1 Table. Annotated genomes of "*Ca*. B. cookevillensis" (CC99) and "*Ca*. B. aquae" (HT99).**
(XLSX)

**S2 Table. Genes involved in amino acid biosynthesis reactions.**
(DOCX)

**S3 Table. Comparison of "*Ca*. B. cookevillensis" (CC99) and "*Ca*. B. aquae" (HT99) dot/icm genes with *L. pnemophilia* dot/icm genes.**
(XLSX)

**S4 Table. Genes encoding putative effector proteins in "*Ca*. B. cookevillensis" (CC99) and "*Ca*. B. aquae" (HT99).**
(XLSX)

**S1 Text. Amino acid biosynthesis pathways of "*Ca*. B. cookevillensis" (CC99) and "*Ca*. B. aquae" (HT99).**
(DOCX)

## Acknowledgments

The authors of this manuscript would like to acknowledge Joyce Miller of the MIMIC Center of MTSU for technical assistance with electron micrographs.

## Author Contributions

**Conceptualization:** Destaalem T. Kidane, John H. Gunderson, Anthony L. Farone, Mary B. Farone.

**Data curation:** Destaalem T. Kidane.

**Formal analysis:** Destaalem T. Kidane, Brock A. Arivett.

**Funding acquisition:** Mary B. Farone.

**Investigation:** Destaalem T. Kidane, Yohannes T. Mehari, Forest C. Rice, Mary B. Farone.

**Methodology:** Destaalem T. Kidane, Mary B. Farone.

**Project administration:** Mary B. Farone.

**Resources:** Mary B. Farone.

**Supervision:** Brock A. Arivett, Mary B. Farone.

**Validation:** Destaalem T. Kidane.

**Visualization:** Destaalem T. Kidane, Forest C. Rice.

**Writing – original draft:** Destaalem T. Kidane.

**Writing – review & editing:** Destaalem T. Kidane, Yohannes T. Mehari, Forest C. Rice, Brock A. Arivett, John H. Gunderson, Anthony L. Farone, Mary B. Farone.

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
