## [Decision Letter · Decision Letter 0]

26 Sep 2022

PONE-D-22-22040The inside scoop: comparative genomics of two intranuclear bacteria, “Candidatus Berkiella cookevillensis” and “Candidatus Berkiella aquae”PLOS ONE

Dear Dr. Farone,

Thank you for submitting your manuscript to PLOS ONE. After careful consideration, we feel that it has merit but does not fully meet PLOS ONE’s publication criteria as it currently stands. Therefore, we invite you to submit a revised version of the manuscript that addresses the minor points raised during the review process.

We look forward to receiving your revised manuscript.

Kind regards,

Daniel E. Voth, Ph.D.

Academic Editor

PLOS ONE

Journal Requirements:

3. Please include your tables as part of your main manuscript and remove the individual files. Please note that supplementary tables should remain as separate "supporting information" files.

Reviewers' comments:

Reviewer's Responses to Questions

**Comments to the Author**

1. Is the manuscript technically sound, and do the data support the conclusions?

Reviewer #1: Yes

Reviewer #2: Yes

2. Has the statistical analysis been performed appropriately and rigorously? 

Reviewer #1: N/A

Reviewer #2: Yes

3. Have the authors made all data underlying the findings in their manuscript fully available?

Reviewer #1: Yes

Reviewer #2: Yes

4. Is the manuscript presented in an intelligible fashion and written in standard English?

Reviewer #1: Yes

Reviewer #2: Yes

5. Review Comments to the Author

Reviewer #1: This is an interesting paper describing the genomic features of two Legionella-like bacteria. The authors have done a thorough analysis of genes and metabolic pathways present in the two Berkiella species. My comments are below:

Line 67: I am not sure whether Berkiella actually belongs in Coxiellaceae family. This taxonomic classification was based on just three genes (16S, rpoS and mip), but more recent trees based on >100 proteins placed Berkiella outside of families Legionellaceae and Coxiellaceae (PMID: 34009306). The authors should either perform a more thorough phylogenetic analysis or mention that the current taxonomic classification is provisional.

L219 and others listed below: Comparison of all genes/pathways present in Berkiella to Legionella and Coxiella would be useful in better understanding Berkiella’s biology and virulence potential. In addition, it could help clarify the evolutionary relationship between the three bacteria.

L219: is glucokinase (glk) missing in Legionella as well?

L241: is the ED pathway present in Coxiella?

L446: Are Tfp genes present/absent in Coxiella?

L470, 478: Are TPR and STPK proteins present in Coxiella?

L504: is relA/sopT present in Coxiella?

L532 and L73: Were the authors able to gain any insights into why CC99 but not HT99 is able to infect mammalian cells?

Table 2: “HT99” should be moved to the cell on the right side.

Fig. 6. Switch A and B.

Reviewer #2: After minor editing and revision, this article is a great value to the literature on comparative genomics of intracellular bacteria. The results presented are original and impacts on the understanding of adaptation of a bacterium to its environment, particularly to the nucleus. This paper is technically sound and adhere to the field standards for experimentation, nomenclature and public availability of data. I officially recommend to accept this paper for publication in PLoS ONE. It would be of interest to see what will be the readers peer review for this article.

6. PLOS authors have the option to publish the peer review history of their article (what does this mean?). If published, this will include your full peer review and any attached files.

Reviewer #1: No

Reviewer #2: **Yes: **Damien F Meyer

---

## [Author Response · Author response to Decision Letter 0]

10 Nov 2022

Dear Dr. Voth,

Thank you for the opportunity to submit a revised draft of our manuscript. We appreciate the work of the reviewers and their positive and constructive comments. We have submitted the documents but want to note that we were unable to upload the Manuscript with Track Changes and the Reviewer Response at the beginning of the documents list. Thus, they appear at the end of the documents.

We have incorporated the suggestions made by the reviewers and have outlined those changes as follows:

Reviewer #1

This is an interesting paper describing the genomic features of two Legionella-like bacteria. The authors have done a thorough analysis of genes and metabolic pathways present in the two Berkiella species. My comments are below:

We thank the reviewer for the encouraging overview of our manuscript.

Line 67: I am not sure whether Berkiella actually belongs in Coxiellaceae family. This taxonomic classification was based on just three genes (16S, rpoS and mip), but more recent trees based on >100 proteins placed Berkiella outside of families Legionellaceae and Coxiellaceae (PMID: 34009306). The authors should either perform a more thorough phylogenetic analysis or mention that the current taxonomic classification is provisional.

We are aware of this and other trees that have placed Candidatus Berkiella spp. outside of the Coxiellaceae, and we thank the reviewer for their resourceful interest in the phylogenetics. We are actually quite excited that others have taken interest in these bacteria. When we performed our initial analyses, we had the understanding that this placement might be provisional due to the lack of sequences for similar bacteria, and that our initial phylogenetic analyses in 2015 were based on sequencing using an Illumina MiSeq platform which generated multiple (40-55) contigs. Our genomic sequencing on which this manuscript was based used a PacBio instrument which produced high quality long reads. We are in the process of revising our phylogenetic analyses as well, and with the publication of this manuscript, this data will also be available to others to generate improved phylogenetic trees. Thus, for this manuscript revision, we have addressed the reviewer’s concern by acknowledging the tree that the reviewer references: LL68-70.

L219 and others listed below: Comparison of all genes/pathways present in Berkiella to Legionella and Coxiella would be useful in better understanding Berkiella’s biology and virulence potential. In addition, it could help clarify the evolutionary relationship between the three bacteria.

We agree with the reviewer as to the usefulness of these comparisons and have not only included the comparisons listed below but have also revised Figure 3 such that both C. burnetii and L. pneumophila are compared along with CC99 and HT99 (previously the figure only compared CC99 and HT99).

L219: is glucokinase (glk) missing in Legionella as well? 

L. pneumophila has a complete glycolytic pathway (Fig 3, L224)

L241: is the ED pathway present in Coxiella? 

No. We have amended the document stating this: LL245-246

L446: Are Tfp genes present/absent in Coxiella?

C. burnetti does not have a complete set of genes for functional Tfp pili. We have amended the manuscript: LL451-452 

L470, 478: Are TPR and STPK proteins present in Coxiella?

Yes. C. burnetii also has TPR and STPK proteins. We have included this information: L480 and L492

L504: is relA/sopT present in Coxiella?

Yes. C. burnetii also has relA/sopT and we have amended the manuscript to include this: LL513-514.

L532 and L73: Were the authors able to gain any insights into why CC99 but not HT99 is able to infect mammalian cells?

In LL543-549, we have included our response to this question. These analyses focused on potential virulence factors and metabolism. We were hoping that these analyses might provide more insight, other than differences in amino acid auxotrophy and folate synthesis. We had added that, other studies (RNAseq of hosts/bacteria and membrane trafficking) are ongoing in our lab that will help to provide more information for this difference in host specificity. 

Table 2: “HT99” should be moved to the cell on the right side.

Thank you. This has been completed.

Fig. 6. Switch A and B.

This has been completed.

Reviewer #2

After minor editing and revision, this article is a great value to the literature on comparative genomics of intracellular bacteria. The results presented are original and impacts on the understanding of adaptation of a bacterium to its environment, particularly to the nucleus. This paper is technically sound and adhere to the field standards for experimentation, nomenclature and public availability of data. I officially recommend to accept this paper for publication in PLoS ONE. It would be of interest to see what will be the readers peer review for this article.

The authors would like to thank Dr. Meyer for his encouraging remarks as to the value of our manuscript. We also look forward to the readers’ response should our manuscript be accepted for publication.

Although Dr. Meyer did not provide specifics as to the “minor editing and revision,” in addition to making changes suggested by Reviewer #1, we have made some minor edits (spelling, removing extra spaces, punctuation with commas) that are visible in our Track Changes version of the manuscript. We have also made edits to capitalization in our References.

We are eager to share our results with the scientific community and we hope that we have thoroughly addressed the reviewers’ comments. We look forward to a response from you as to the publication of our manuscript.

Sincerely,

Mary B. Farone, Ph.D.

Professor, Biology Department

Molecular Biosciences Program

Middle Tennessee State University

Murfreesboro, TN 37132

mary.farone@mtsu.edu

615.904.8341

---

## [Editor Report · Decision Letter 1]

14 Nov 2022

The inside scoop: comparative genomics of two intranuclear bacteria, “Candidatus Berkiella cookevillensis” and “Candidatus Berkiella aquae”

PONE-D-22-22040R1

Dear Dr. Farone,

We’re pleased to inform you that your manuscript has been judged scientifically suitable for publication and will be formally accepted for publication once it meets all outstanding technical requirements.

Kind regards,

Daniel E. Voth, Ph.D.

Academic Editor

PLOS ONE
---

## [Editor Report · Acceptance letter]

19 Dec 2022

PONE-D-22-22040R1 

The inside scoop: comparative genomics of two intranuclear bacteria, *“Candidatus Berkiella cookevillensis”* and *“Candidatus Berkiella aquae”*

Dear Dr. Farone:

I'm pleased to inform you that your manuscript has been deemed suitable for publication in PLOS ONE. Congratulations! Your manuscript is now with our production department. 

Kind regards, 

on behalf of

Dr. Daniel E. Voth 

Academic Editor

PLOS ONE